# Enhanced Antisense Oligonucleotide Delivery Using Cationic Liposomes Grafted with Trastuzumab: A Proof-of-Concept Study in Prostate Cancer

**DOI:** 10.3390/pharmaceutics12121166

**Published:** 2020-11-29

**Authors:** Guillaume Sicard, Clément Paris, Sarah Giacometti, Anne Rodallec, Joseph Ciccolini, Palma Rocchi, Raphaëlle Fanciullino

**Affiliations:** 1SMARTc Unit, CRCM Inserm U1068, Aix Marseille University, 13007 Marseille, France; guillaume.sicard@univ-amu.fr (G.S.); sarah.giacometti@univ-amu.fr (S.G.); anne.rodallec@univ-amu.fr (A.R.); joseph.ciccolini@univ-amu.fr (J.C.); raphaelle.fanciullino@univ-amu.fr (R.F.); 2CNRS, INSERM, Institut Paoli-Calmettes, CRCM, Aix Marseille University, 13007 Marseille, France; clement.paris@inserm.fr

**Keywords:** liposomes, immunoliposomes, antisense oligonucleotides, prostate cancer

## Abstract

Prostate cancer (PCa) is the second most common cancer in men worldwide and the fifth leading cause of death by cancer. The overexpression of TCTP protein plays an important role in castration resistance. Over the last decade, antisense technology has emerged as a rising strategy in oncology. Using antisense oligonucleotide (ASO) to silence TCTP protein is a promising therapeutic option—however, the pharmacokinetics of ASO does not always meet the requirements of proper delivery to the tumor site. In this context, developing drug delivery systems is an attractive strategy for improving the efficacy of ASO directed against TCTP. The liposome should protect and deliver ASO at the intracellular level in order to be effective. In addition, because prostate cancer cells express Her2, using an anti-Her2 targeting antibody will increase the affinity of the liposome for the cell and optimize the intratumoral penetration of the ASO, thus improving efficacy. Here, we have designed and developed pegylated liposomes and Her2-targeting immunoliposomes. Mean diameter was below 200 nm, thus ensuring proper enhanced permeation and retention (EPR) effect. Encapsulation rate for ASO was about 40%. Using human PC-3 prostate cancer cells as a canonical model, free ASO and ASO encapsulated into either liposomes or anti-Her2 immunoliposomes were tested for efficacy in vitro using 2D and 3D spheroid models. While the encapsulated forms of ASO were always more effective than free ASO, we observed differences in efficacy of encapsulated ASO. For short exposure times (i.e., 4 h) ASO liposomes (ASO-Li) were more effective than ASO-immunoliposomes (ASO-iLi). Conversely, for longer exposure times, ASO-iLi performed better than ASO-Li. This pilot study demonstrates that it is possible to encapsulate ASO into liposomes and to yield antiproliferative efficacy against PCa. Importantly, despite mild Her2 expression in this PC-3 model, using a surface mAb as targeting agent provides further efficacy, especially when exposure is longer. Overall, the development of third-generation ASO-iLi should help to take advantage of the expression of Her2 by prostate cancer cells in order to allow greater specificity of action in vivo and thus a gain in efficacy.

## 1. Introduction

Prostate cancer (PCa) is the second most common cancer in men worldwide with 1.3 million new cases in 2018. PCa is the fifth leading cause of death by cancer with more than 360,000 deaths in 2018, despite a decrease in its incidence by 6% over 2005–2009 [1,2]. At initial diagnosis, treatment depends on the stage based on Gleason score, the patient’s characteristics and the PSA level [3,4]. Surgical treatment, i.e., prostatectomy, is the standard of care, but patients with advanced disease (i.e., stages III or IV, high Gleason score) are precluded. The first-line therapy for advanced PCa is castration therapy which consists in androgen deprivation since it is a hormone-sensitive cancer [5]. After a period of therapeutic response, usually 1–3 years, patients will ultimately become resistant to the therapy and develop metastases. A new approach is therefore needed for these castration-resistant PCa (CRPCa) patients.

The overexpression of the TCTP protein plays an important role in PCa and most particularly in CRPCa [6]. Indeed, this protein is involved in progression of the disease and therapeutic failure. The interactions between TCTP and p53 and their negative feedback regulation loop are responsible for progression and invasion of PCa [7,8]. Recently, antisense technology has emerged as a promising strategy in cancer [9]. The principle of this approach is the sequence-specific binding of an antisense oligonucleotide (ASO) to target mRNA, thus preventing gene translation [10]. The development of an ASO directed against TCTP seems therefore to be an interesting strategy [11]. The shutting down of TCTP by ASO is expected to restore apoptosis and sensitivity to hormone-therapy and chemotherapy of cancer cells. ASOs are only active after cell uptake, and therefore, a carrier is necessary to help them pass the membranes. In this context, developing carriers to transport ASO is an attractive strategy, especially since nanoparticles are increasingly considered to stretch the efficacy/toxicity balance of a variety of anticancer agents or payloads [12,13,14]. Various other technological approaches such as the direct pegylation of compounds [15] or using Nab conjugates [16] or antibody–drug conjugates [17,18] illustrate this major trend to develop drug carriers in oncology today. One of the critical points when developing nanoparticles is based on their size being <200 nm. Nanoparticles leave the vascular compartment and accumulate in the interstitial space next to the tumor. This phenomenon is called passive targeting or the enhanced permeability and retention (EPR) effect [19].

Our study is based upon this double trend of encapsulation and use of therapeutic monoclonal antibodies to improve the specificity of nanoparticles against tumors. The conjunction of these two concepts results in a new nanoparticle, the antibody nanoconjugate (ANC), more commonly called the immunoliposome [20,21]. In this study, we present the early development steps of an innovative stealth liposomal ASO nanoparticle targeting prostate cancer through anti-Her2 functionalization [22] (Figure 1).

## 2. Materials and Methods

### 2.1. Cell Lines

Experiments were carried on canonical human prostate cancer cell line PC-3 (American Type Culture Collection, Rockville, MD, USA) Cells were cultured in RPMI supplemented with 10% FBS, 1% penicillin and 0.16% kanamycin and grown in a humidified 5% CO_2_ incubator at 37 °C. Cells were regularly authenticated in terms of cell viability, morphology and doubling time. For Her2 characterization, breast cancer cell lines, i.e., MDA231, MDA 453 and SKBR3, were used (American Type Culture Collection, Manassas, VA, USA).

### 2.2. Drugs and Chemicals

1,2-distearoyl-*sn*-glycero-3-phosphoethanolamine-N-(aleimide(polyethyleneglycol)-2000) (Mal-PEG) and 1,2-dioleoyl-3-trimethylammoniumpropane (DOTAP) were purchased from COGER (Paris, France). Egg yolk phosphatidylcholine (PC) and cholesterol (Chol) were purchased from Sigma (St-Quentin-Fallavier, France). ASO was purchased from Eurofins (Les Ullis, France). 2-iminothiolane (Traut’s reagent) and Draq5 were purchased from Fisher Scientific (Illkirch-Graffenstaden, France). QuantiBRITE phycoerythrin (PE) and PE Mouse Anti-Human Her-2/neu were purchased from BD Biosciences (San Jose, CA, USA). Trastuzumab (Herceptin) was kindly given by Genentech (South San Francisco, CA, USA). All other reagents were of analytical grade.

### 2.3. ASO Stability in Solvents

ASO stability in three solvents, i.e., NaCl 0.9%, water and methanol, was tested over one month.

HPLC detection was performed on an HPLC (Agilent 1260, Agilent, Les Ulis, France). The HPLC column Xbrige OST C18 2.5 μm 4.6 × 50 mm (Waters, Guyancourt, France) was equilibrated at a flow rate of 0.8 mL/min. Eluant A contained 0.1 M TEAA (Triethylammonium acetate) in 5% ACN (acetonitrile) in water and eluant B contained ACN. The elution gradient was 0 to 45% in 10 min. ASO was detected at the wavelength of 260 nm. Ninety microliters of ASO in NaCl 0.9% and ASO in water were directly injected. For ASO in MeOH, 100 µL of sample was evaporated to dryness; the sample was resuspended in water, and then 90 µL was injected.

### 2.4. Pegylated Liposome Preparation

Two different compositions of liposomes were studied: formulation 1, using DOTAP, Mal-PEG and Chol, and formulation 2, using DOTAP, PC, Mal-PEG and Chol.

Both compositions were prepared using the classic thin-film method [23]. Briefly, lipids were dissolved in methanol as organic solvent. Methanol was then removed by rotary evaporation (Laborota 4003, Heidolph Instruments, Schwabach, Germany) at 38 °C under vacuum to avoid further toxicity. After 30 min, a thin lipid film was obtained. To remove the residual solvent, lipid film was dried under a stream of nitrogen for 2 h at room temperature. The film was then hydrated with a 5% *v*/*v* dextrose solution in water for formulation 1 or a 0.9% *v*/*v* sodium chloride solution in water for formulation 2, and then multilamellar vesicle (MLV) liposomes were obtained. Extrusion step was performed to reduce and homogenize liposomes in size through two 0.1 µm and two 0.08 µm polycarbonate pore membranes (Nucleopore, Whatman, Maidstone, UK) using LipoFast LF-50, and then small unilamellar vesicle (SUV) liposomes were obtained [24].

For each formulation, liposomes (i.e., ASO-Li-1 for formulation 1 and ASO-Li-1 for formulation 2) and immunoliposomes (i.e., ASO-iLi-1 and ASO-iLi-2) were generated.

### 2.5. Pegylated Immunoliposome Preparation: Encapsulation Strategies

Different encapsulation strategies were tested with both formulations.

With formulation 1, ASO was introducing in methanol with lipids during the thin lipid film formation. Extraction from liposome was performed by 100 µL methanol for 100 µL liposome.

With formulation 2, three different strategies were tested. First, ASO was included in aqueous solvent, i.e., sodium chloride 0.9%, during the hydration step of the lipidic film. ASO was also included in preformed liposomes by contact using other strategies, namely soft agitation by rotation at 38 °C or fast agitation using bar magnet at 38 °C.

Besides, a new extraction technique was used with a chloroform–methanol solution (1:2 ratio) [25]. For each 100 µL of liposome solution, we added 375 µL of chloroform–methanol solution. Then, 125 µL of chloroform was added to the sample and vortexed. Afterward, 125 µL of MilliQ water was added to the sample and vortexed again. Finally, the sample was centrifugated at 1500× *g* for 90 s. The topper phase contained the ASO freed from the lipids. The entire aqueous phase was recuperated, evaporated and reconstituted with 100 µL of MilliQ water.

### 2.6. Pegylated Immunoliposome Preparation: Antibody Engraftment

Trastuzumab engraftment was performed from previously obtained liposome, generating immunoliposome.

The engraftment was carried out by maleimide–thiol conjugation after having previously derivatized the trastuzumab with thiol function. To this end, trastuzumab was first dissolved in a 0.1 M sodium phosphate-buffered saline (PBS), pH 8.0, containing 5 mM ethylene diamine tetra-acetic acid, and mixed under constant shaking for 2 h at room temperature with a Traut’s reagent solution at 1:10 molar ratio (Traut’s/trastuzumab). Thiolated trastuzumab was then directly mixed with the pegylated liposomes at 1:127 molar ratio (trastuzumab/MAL-PEG).

The mixture was kept under constant shaking at 4 °C overnight. Unbound trastuzumab and free ASO were removed from the liposomal solution using 6000× *g* centrifugation on MWCO 300 KDa Vivaspin (VWR, Fontenay-sous-Bois, France) [26].

### 2.7. Size and Polydispersity Study

Size and polydispersity index (PDI) of liposomes and immunoliposomes were measured by dynamic light scattering (DLS). Analysis parameters were as follows: medium: PBS solution, viscosity of water: 0.8937, temperature: 25 °C, dielectric constant: negative but not used in these measurements, nanoparticles: liposomes, refractive index of water: 1.333 cP, detection angle: 173°, wavelength: 632.8 nm, software for analysis of data: Zetasizer Nano software v3.30.

Liposomes and immunoliposomes were diluted in a PBS solution and then analyzed by a Zetasizer Nano S (Malvern Instruments, Malvern, UK). Liposomal preparations were considered unimodal for a PDI < 0.2. The measures were performed extemporaneously after liposome formation for both formulations in triplicate.

Stability study was performed at 7, 14 and 28 days after liposome formation.

### 2.8. ASO Encapsulation Rate

ASO encapsulation rate was only determined for formulation 2 by fluorescence (495/520 nm). After grafting ASO with FITC in 3′ (Eurofins, Les Ullis, France), ASO–FITC encapsulation rate was determined by measuring the FITC fluorescence at 520 nm (PHERAstar FSX, BMG Labtech, Ortenberg, Germany) [27]. All the measures were performed in triplicate.

### 2.9. Quantification of HER2 on Cells

Flow cytometry analysis allowed measuring the expression of Her2 on the surface of PC-3 cells [28].

As previously described [29], QuantiBRITE PE (BD Biosciences, San Jose, USA) was used to estimate the absolute number of Her2 receptors on cell membranes. About 100,000 cells of PC-3 were incubated under saturated conditions with PE Mouse Anti-Human HER-2/neu (BD Biosciences, San Jose, USA) for 30 min at 4 °C before being rinsed with PBS. IgG2a-PE anti-mouse antibodies (Fisher Scientific, Illkirch, France) were used for isotopic control. Analysis was then immediately performed on Gallios Beckman Coulter. Assuming our anti-HER-2 PE antibody has a 1:1 fluorochrome/antibody ratio, PE median fluorescence intensity (MFI) was measured for all cell lines and reported on a log–log graph with MFI vs. PE molecules, after subtracting isotopic control MFI. All the measures were performed in triplicate.

### 2.10. In Vitro Assays

Spheroids were obtained with PC-3 cells seeded with 20% methylcellulose solution on a 96-well U-bottom plate for at least 24 h before the experiment began. Different drug concentrations and scheduling conditions were tested on 5000-cell spheroids. Viability was assessed by bioluminescence assay. The cell viability in bioluminescence was determined on PC-3 cells using luminescent cell viability assay (CellTiter-Glo, Promega, Madison, WI, USA) and bioluminescence reading (PHERAstar FSX; BMG Labtech, Ortenberg, Germany). Cellular uptake was observed using confocal microscopy (TCS SP2 Leica) coupled to a digital camera.

Using 5000-cell spheroids, we determined nontoxic concentrations of lipid treated on day 1 and day 8 with viability determined in bioluminescence at day 15.

Subsequently, in a first protocol, we tested one nontoxic concentration of lipid (i.e., 8 nM) following day 3 and day 10 (after spheroid formation) treatment schedule with viability assay on day 15 or only at day 3 with viability assay at day 15. Encapsulated ASO concentration (i.e., after lipidic film formation) at 150 nM was tested. Cells were exposed to treatment for four hours on Day 3 and/or day 10. After four hours of exposure, treatment was removed and replaced by supplemented RPMI.

In a second protocol, using 5000-cell spheroids, we tested two nontoxic concentrations of lipid (i.e., 2 and 8 nM) following day 3 and day 10 treatment schedule with viability assay on day 15. Encapsulated ASO concentration (i.e., after lipidic film formation) at 150 nM was tested. Empty liposomes and immunoliposomes, for each nontoxic lipid concentration, were used as control.

### 2.11. Statistical Analysis

Formulation experiments were performed at least in triplicate and data were represented as mean ± SD or ±standard error of the mean. Statistical analyses were performed on MedCalc 17.2.1. Software (MedCalc, Acacialaan, Belgium).

In vitro experiment was performed at least in triplicate and data were represented as mean ± SD or ±standard error of the mean. All statistical analyses were performed with car [30] and multcomp [31] packages of the software R [32].

## 3. Results

### 3.1. ASO Stability in Solvents

ASO stability has been studied in three different solvents. Sodium chloride 0.9% solution was tested because it was used during the hydration step of the lipidic film during liposome formation, water was used because lyophilized ASO was reconstituted with it and methanol was the organic solvent used for lipid dissolution before lipid film formation and for liposome extraction.

Results show that ASO is stable in the three solvents over 32 days. Thus, formulation tests can be carried out by including ASO in the organic phase (methanol) or in the hydration solvent (NaCl 0.9%) without fear of deterioration of the ASO (Figure 2).

### 3.2. Size and Polydispersity

We have developed two lipidic compositions: DOTAP/Mal-PEG/Chol (29:2:69) and DOTAP/Mal-PEG/PC/Chol (20:20:58:2) respectively for ASO-Li-1/ASO-iLi-1 and ASO-Li-2/ASO-iLi-2.

Size and PDI are summarized in Table 1.

We observed a statistically significant difference in size between ASO-Li-1 and ASO-iLi-1 (*p* = 0.016) and ASO-Li-2 and ASO-iLi-2 (*p* = 0.032, Student’s *t*-test).

Stability study was performed on days 7, 14 and 28 after liposome formation. For formulation 1, size showed an increase both for ASO-Li-1 (*p* = 0.007, Student’s *t*-test) and ASO-iLi-1 (*p* = 0.002, Student’s *t*-test). Given these results, stability study was not performed on days 14 and 28.

For formulation 2, size showed no difference from first measures (*p* = 0.838 for ASO-Li-2 and *p* = 0.838 for ASO-iLi-2, one-way ANOVA testing). Results for size and PDI monitoring over time are summarized in Table 2.

### 3.3. Pegylated Immunoliposome Preparation: Encapsulation Strategies

ASO–FITC was encapsulated into liposomes composed of DOTAP/Mal-PEG/PC/Chol in the molar ratio 20:20:58:2. Mean encapsulation rates of ASO–FITC were 21.14 ± 7.65% for fast agitation, 42.88 ± 3.80% for soft agitation and 39.00 ± 2.16% combined hydration solvent and soft agitation (Figure 3). No significant difference appears between the two formulations using soft agitation *p* = 0.484). Whether ASO was introduced in organic or in hydrophilic phase had no influence on ASO encapsulation.

### 3.4. Quantification of Her2 on Cells

PC-3 cells were found to express 12 × 10^3^ ± 0.5 × 10^3^ Her2 receptors per cell. This was next compared to the expression of Her2 receptor on breast cancer cells MDA-MB-231, MDA-MB-453 and SKBR3 (reference cells for Her2 expression), as shown in Figure 4.

### 3.5. 3D In Vitro

#### 3.5.1. Lipidic Antiproliferative Activity

For both liposomes and immunoliposomes and for both formulations, the nontoxic concentration in empty liposomes was observed for lipid concentrations <10 nM for PC-3 cells treated at days 1 and 8 with viability determined in bioluminescence at day 15 (Figure 5). These results agree with those obtained in 2D (data not shown).

Following these results, we decided to work with formulation 2 at two nontoxic lipid concentrations (2 and 8 nM) and an encapsulated ASO at 150 nM. Encapsulation rate for each batch was 41 ± 6%.

#### 3.5.2. Liposomal Antiproliferative Activity

All the measures were performed in triplicate; these protocols (protocols 1 and 2) are summarized in Figure 6 and Figure 7.

Empty liposomes and immunoliposomes showed no in vitro cytotoxicity (data not shown). Results of cytotoxic studies with protocol 1 are summarized in Figure 8. Greater cytotoxicity was observed in PCa exposed to liposomes with both treatment schemes. No effect was detected with ASO-iLi-2 at 8 nM with both schedules.

Results of cytotoxic studies with protocol 2 at low lipid concentration are summarized in Figure 9. Greater cytotoxicity was observed in PCa exposed to immunoliposomes encapsulated with 150 nM of ASO and to lipid concentration of 2 nM (*p* = 0.039). No cytotoxic effect on PCa cells was detected after ASO-Li-2 treatment at low lipid concentration.

Results of cytotoxic studies with protocol 2 at high lipid concentration are summarized in Figure 10. A greater cytotoxicity was observed in PCa exposed to immunoliposomes encapsulated 150 nM of ASO (*p* = 0.0041 vs. Empty-iL-2 and *p* = 0.0039 vs. ASO-Li-2).

Cell viability results obtained were confirmed by microscopic observation (Figure 11). Observations were made on days 1 (24 h after cell seeding), 8 and 15.

Microscopy observations were consistent with bioluminescence ones. On day 15, spheroids treated with ASO-iLi-2 at 8 nM seemed to present more cellular death than control and other treatment conditions.

## 4. Discussion

ASOs are a promising approach in oncology, and more particularly in CRPCa with the development of ASO targeting TCTP.

While ASO has shown its efficacy in vitro, cellular uptake has proven to be much more challenging. In this respect, we have here studied a new encapsulated formulation of ASO, designed to penetrate in cells. In this work, we have developed innovative second- and third-generation liposomal formulations encapsulating ASO, so as to enhance cellular penetration and effectiveness in PCa cells [33,34,35,36].

Pegylated immunoliposomes, thanks to their stealthiness and size compatible with the EPR effect, provide a unique opportunity for both passive and active targeting of tumors. To this end, liposomal formulations must take into account two important parameters: the nature of the ASO (i.e., its hydrophilic nature and its anionic character) and the necessity of a diameter below 200 nm for EPR effect. Here, anti-Her2 trastuzumab was used as a targeting agent because Her2 is frequently expressed in several cancers such as breast cancer or gastric/gastroesophageal cancers [37].

We found that the level of Her2 expression in PC-3 was relatively low, so previous work on MDA-MB231 cells with even a lower level of target expression than PC-3 showed promising results in our laboratory. Still, the rationale for targeting Her2 on PC-3 cells could be questioned here. Because the PC-3 cells are the cells mostly used as an experimental model for CRPa [38,39], they were considered as a fully suitable model for evaluating the usefulness of transporting ASOs with immunoliposomes in prostate cancer, especially because we have demonstrated previously that higher efficacy with trastuzumab-grafted liposomes was not necessarily correlated to a high level of Her2 expression.

The innovative nature of this approach consisted in the combination of the advantages of a small diameter (i.e., <150 nm) of lipid–nucleic acid nanoparticles, improving ASO drug delivery through passive targeting, with the advantages of active targeting (i.e., anti-Her2) [40,41,42]. Prostate tumors are highly vascularized tumors, thus making this tumor eligible for EPR effects [43,44].

Previous studies have highlighted the complexity of defining optimal formulations yielding acceptable encapsulation rates with other hydrophilic active agents such as siRNA [45]. In our study, the difficulty was to combine acceptable encapsulation rates with small-sized nanoparticles while allowing trastuzumab engraftment.

In the present study, we showed that is it possible to design ASO liposomes and immunoliposomes. We have developed two different liposomal formulations showing good performances in terms of size, encapsulation rate and stability. DOTAP is a cationic lipid used to optimize liposome stability and increase anionic oligonucleotide ASO encapsulation rate. Chol and PC are neutral lipids frequently used for liposome formulation. The two tested formulations differ by PC presence in order to increase liposome stability. Finally, Mal-PEG is a common stealth agent used for all kinds of nanoparticles [46,47].

We worked with two nontoxic lipid concentrations (2 nM and 8 nM) and an encapsulated ASO at 150 nM. Indeed, lipid concentration plays a major role in liposome stability and leakage from the membrane. Low cholesterol concentration decreases stability and increases drug release [48]. Testing two lipid concentration levels allowed differences in cytotoxic effect to be highlighted.

Moreover, cholesterol affects the plasticity of the liposomal membrane; i.e., increasing cholesterol decreases the plasticity and increases the rigidity of the membranes [49]. Cholesterol was therefore a critical component because, for hydrophilic components such as ASO, membrane rigidity determines the release rate of the content [50].

A significant difference was found between formulations using fast agitation and the two other formulations (*p* < 0.001). This difference can be explained by destabilization and weakening of liposomes during fast agitation.

Following these results, we decided to work with formulation 2 composed of DOTAP/Mal-PEG/PC/Chol in the molar ratio 20:20:58:2. The encapsulation rate for each batch was 41 ± 6%.

Liposomes and immunoliposomes were compared in terms of physical characteristics, size suitable for EPR effect and encapsulation rates. The extraction of the ASO was optimized using a chloroform–methanol mixture. Finally, soft agitation with thin-film method was selected as the best encapsulation method, yielding encapsulation rates of about 50%, i.e., twice as much as the other methods we tested.

However, for the present proof-of-concept study, aiming at demonstrating that encapsulating ASOs in lipidic carriers helps to increase their efficacy, we decided to work on freshly prepared batches, rather than trying to improve shelf-stability.

Antiproliferative assays were performed on PCa-3 cells using both 2D and 3D models. Spheroids (i.e., 3D) are considered as a better model for evaluating the efficacy of nanoparticles [51]; however, 2D models were useful for determining IC50s of free ASO and free trastuzumab, as well as for determining the nontoxic concentrations of blank liposomes. These preliminary results on 2D models helped us to determine the concentrations (i.e., 2 and 8 nM) to be used for the 3D testing without confounding factors such as the direct toxic effect of overly concentrated lipids.

When using 3D models, the antiproliferative activity of free ASO was low and fully in line with that already described in the literature, thus confirming the need to provide ASO with a carrier to further increase its efficacy. Empty liposomes and blank immunoliposomes did not show antiproliferative activity. In 3D culture, the nontoxic concentration in lipids was confirmed to be 10 nM.

For free drugs (i.e., free ASO and free trastuzumab), no efficacy was observed in 3D models (data not shown). These data confirmed that ASO needs to be encapsulated to be efficient. We subsequently developed two processing protocols taking into account the preliminary data available [22].

Regardless of the short exposition time (4 h) and delayed exposure (i.e., day 3 and days 3 and 10), we observed a significant superiority in the efficacy of the liposomes as compared to the immunoliposomes. On the other hand, in these conditions of brief exposure, the treatment of PCa cells by immunoliposomes did not show any efficacy.

This scheduling was previously used in 2D models with PC-3 and already showed an effect in downregulation of TCTP expression and cell viability when using lipid-conjugated ASO carrier [12]. However, with our liposomes and immunoliposomes, these results were not confirmed. This loss of efficacy for immunoliposomes encapsulating ASO could come from this incubation time being too short to allow proper interaction between trastuzumab and Her2 receptors on PC-3 cells, preventing the efficient intracellular release of ASO. Indeed, the presence of the grafted antibody in the bilayer of the nanoparticle causes steric hindrance and is probably accompanied by a longer cellular uptake, resulting in a delayed release of ASO at the cellular level and limiting its efficacy.

In protocol 2 with extended exposure time, no significant differences were found between liposome-encapsulated ASO and empty liposome at all lipid concentrations. We have demonstrated significantly superior efficacy of ASO-iLi-2 at any lipid concentration for long exposure times when compared to liposome treatment.

In addition, for long exposure times, we have demonstrated significantly superior efficacy of ASO-iLi-2 at 8 nM lipid concentration when compared to ASO-Li-2 treatment.

Trastuzumab, therefore, seems to play a critical role in the active targeting of PC-3 cells, despite their low expression level of Her2. Indeed, free trastuzumab showed no efficacy in 3D models. Thus, the greater antiproliferative efficacy achieved with immunoliposomes as compared with liposomes encapsulating ASO cannot be directly related to some kind of direct trastuzumab cytotoxicity. Instead, better targeting of cancer cells by passive and active targeting due to a better distribution of the immunoliposomes in the core of the spheroid could explain this increase in efficacy.

Moreover, the differences in efficacy observed between immunoliposomes used at 2 and 8 nM can be explained by differences in the membrane plasticity of the carriers. Indeed 2 nM immunoliposomes contain less cholesterol than 8 nM immunoliposomes, and consequently, the membrane is less rigid, resulting in a possible leaking process due to poor stability. Conversely, the lipid concentration of 8 nM immunoliposomes allows better protection and subsequently also better intracellular penetration of ASO into cancer cells.

The challenge of this work was to highlight the possibility of encapsulating the ASO into lipidic nanoparticles so as to increase its efficacy. For both the developed liposomes and immunoliposomes, we have demonstrated that lipidic nanoparticles are suitable for encapsulating ASOs in terms of size, short-term stability and encapsulation rates. Results showed that both liposomes and immunoliposomes performed better than free ASO, thus suggesting that encapsulating ASO could help to increase its cytotoxicity. Because nanoparticles do not aim at changing the pharmacology of a drug, but rather aim at affecting the delivery and pharmacokinetics, we hypothesize here that lipids containing ASO could help carry the payload inside tumors.

## 5. Conclusions

In this proof-of-concept study, we have demonstrated that it is possible to increase the efficacy of ASO in the canonical PC-3 model for prostate cancer by using lipid carriers. Interestingly, immunoliposomes targeting Her2 have presented the most promising efficacy on 3D spheroids, provided that incubation time was long enough, despite a low expression of Her2 in PC-3 cells. Conversely, short exposure times led to higher efficacy of liposomes as compared with immunoliposomes. This difference could come from the steric hindrance of the immunoliposomes requiring a delay in the release of ASO, thus making it a more suitable candidate for long-exposure schedules. Overall, this work suggests that shifting from standard intratumoral administration of oligonucleotides to systemic administration is feasible, provided that a suitable vehicle is developed.

Poor stability of encapsulated ASO obliged us to use extemporaneously prepared batches; however, this weakness could be fixed in the future, i.e., by lyophilization of the nanoparticles so as to enhance shelf-stability.

## Figures and Tables

**Figure 1 pharmaceutics-12-01166-f001:**
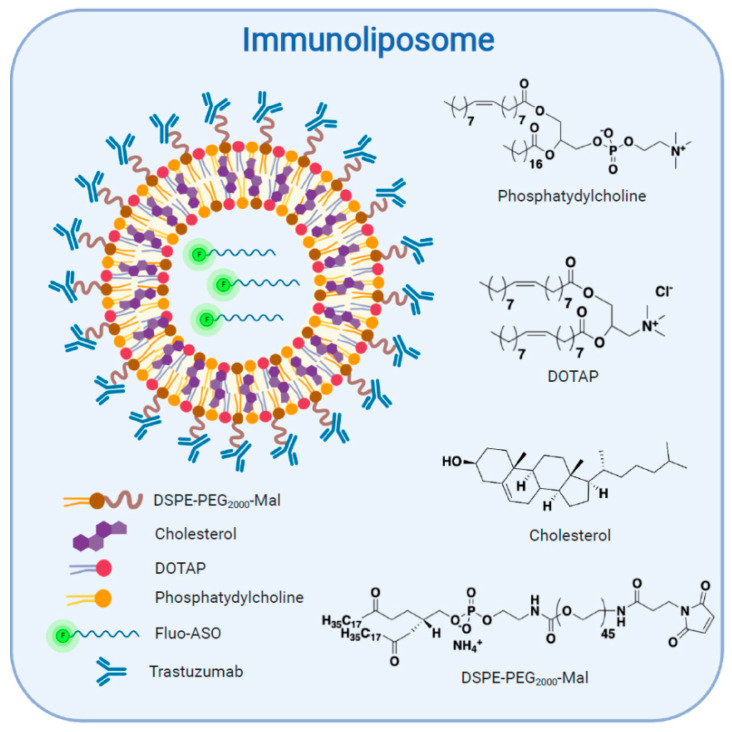
Schematic representation of the immunoliposome structure and composition.

**Figure 2 pharmaceutics-12-01166-f002:**
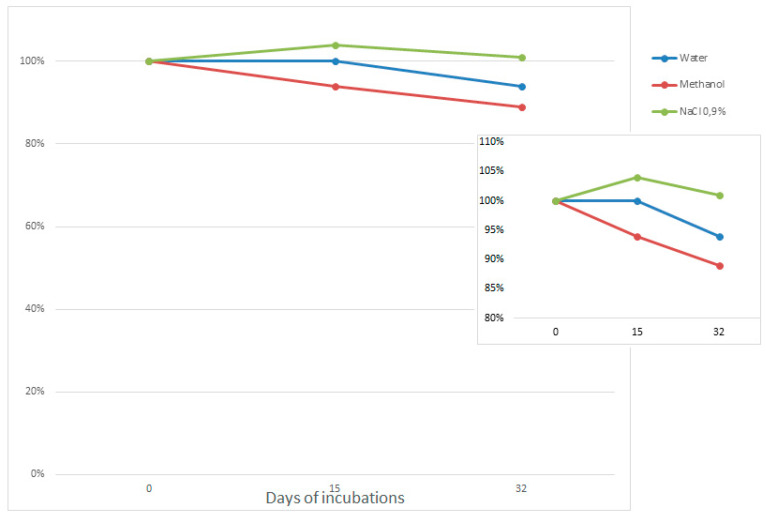
Normalized percentage of full antisense oligonucleotide (ASO) length over time for three solvents.

**Figure 3 pharmaceutics-12-01166-f003:**
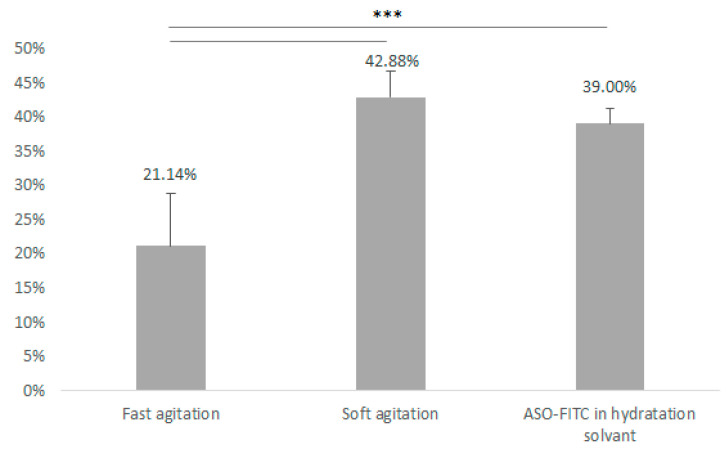
Influence of different strategies on encapsulation rate (***, 0; **, 0.001; * 0.01).

**Figure 4 pharmaceutics-12-01166-f004:**
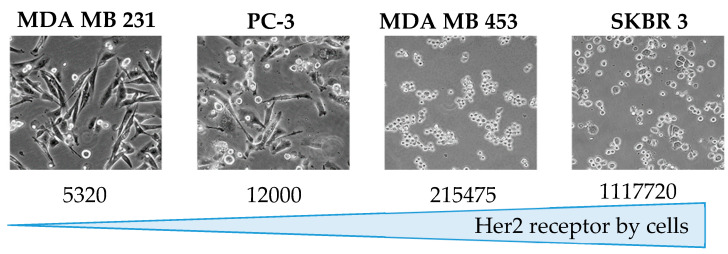
Comparison of Her2 receptors on cell surface for MDA-MB-231, PC-3, MDA-MB-453 and SKBR3.

**Figure 5 pharmaceutics-12-01166-f005:**
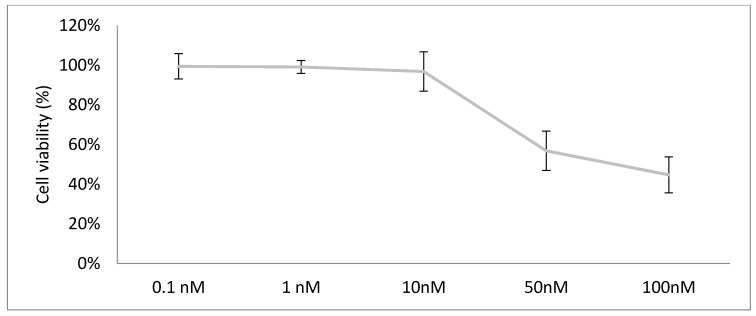
Cell viability (%) of PC-3 when exposed to Empty-Li-2.

**Figure 6 pharmaceutics-12-01166-f006:**
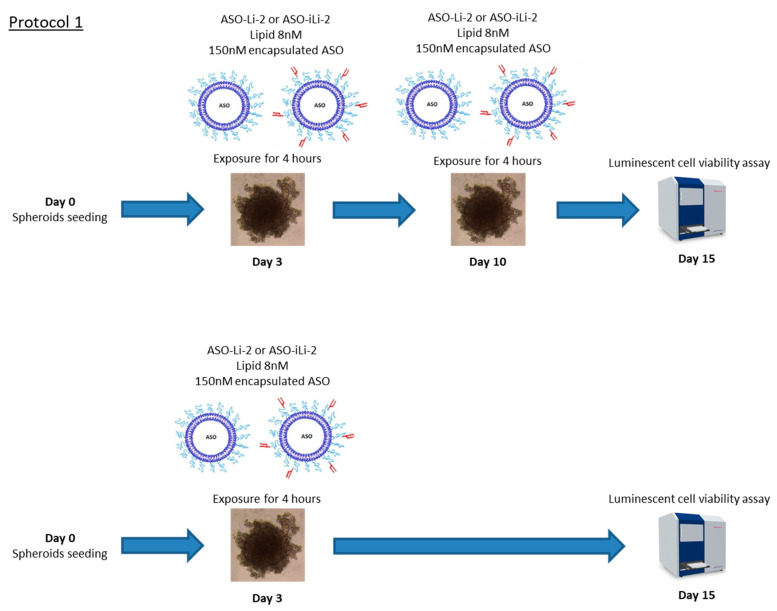
Protocol 1: liposomal antiproliferative activity with short exposition time. 3D in vitro assessment with spheroid treatment protocol 1. All the measures were performed in triplicate.

**Figure 7 pharmaceutics-12-01166-f007:**
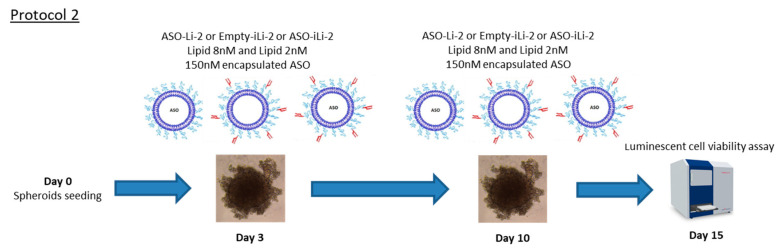
Protocol 2: liposomal antiproliferative activity with long exposition time. 3D in vitro assessment with spheroid treatment protocol 2. All the measures were performed in triplicate.

**Figure 8 pharmaceutics-12-01166-f008:**
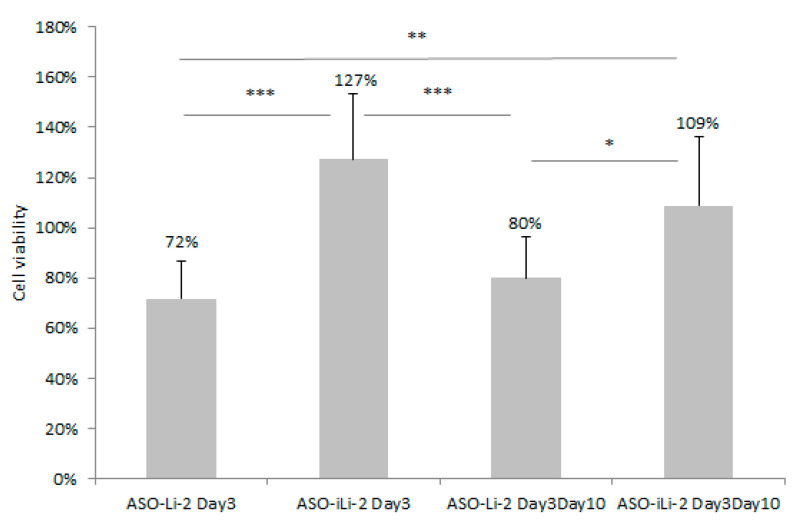
Cell viability (%) of 5000 PC-3 cell spheroids when exposed to ASO-Li-2 at 8 nM and ASO-iLi-2 at 8 nM for 4h at day 3 and days 3–10 (***, 0; **, 0.001; * 0.01).

**Figure 9 pharmaceutics-12-01166-f009:**
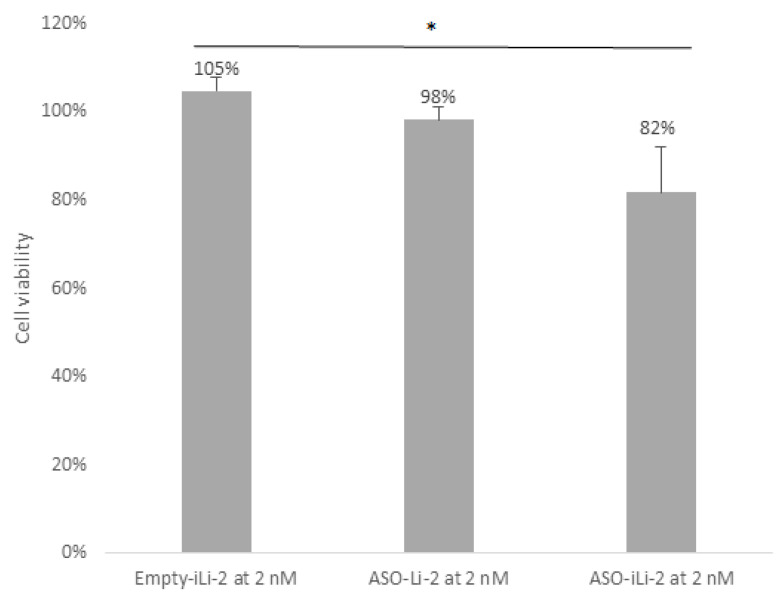
Cell viability (%) of 5000 PC-3 cell spheroids when exposed to Empty-iLi-2 at 2 nM, ASO-Li-2 at 2 nM and ASO-iLi-2 at 2 nM (***, 0; **, 0.001; * 0.01).

**Figure 10 pharmaceutics-12-01166-f010:**
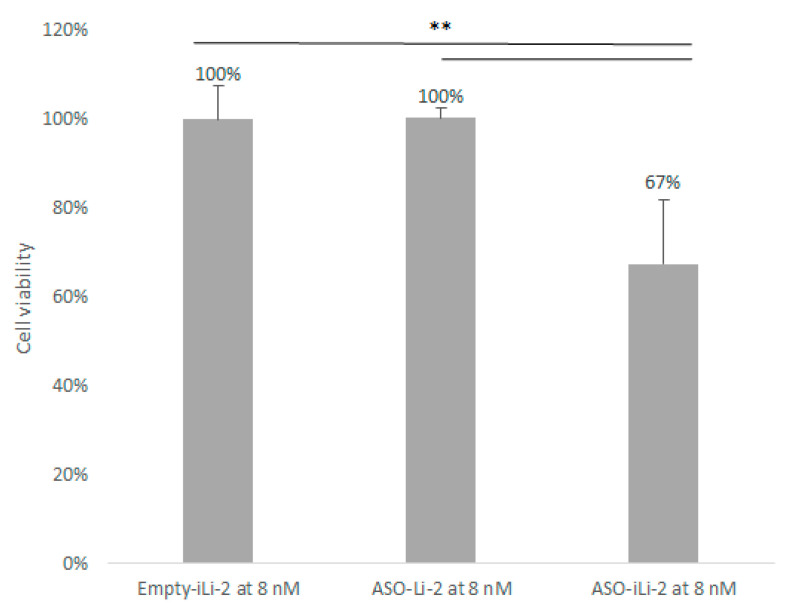
Cell viability (%) of 5000 PC-3 cell spheroids when exposed to Empty-iLi-2 at 8 nM, ASO-Li-2 at 8 nM and ASO-iLi-2 at 8 nM (***, 0; **, 0.001; * 0.01).

**Figure 11 pharmaceutics-12-01166-f011:**
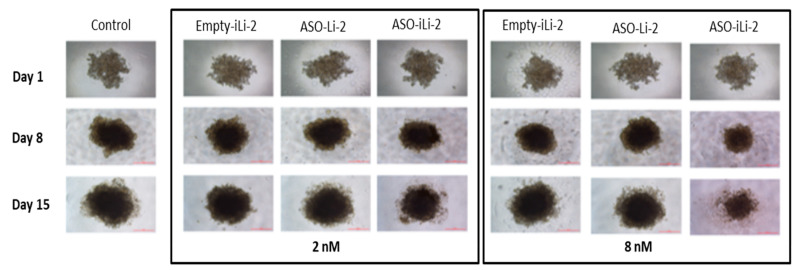
Spheroid (5000 cells) microscopy observations.

**Table 1 pharmaceutics-12-01166-t001:** Size and polydispersity index (PDI) comparison for both formulations of liposomes and immunoliposomes.

Formulation	Size (nm) ± SD	PDI ± SD
**ASO-Li-1**	127.7 ± 4.8	0.161 ± 0.03
**ASO-iLi-1**	139.6 ± 1.8	0.183 ± 0.06
**ASO-Li-2**	145.6 ± 4.1	0.080 ± 0.01
**ASO-iLi-2**	154.1 ± 9.4	0.090 ± 0.03

**Table 2 pharmaceutics-12-01166-t002:** Size and PDI comparison over time for formulation 2.

Formulation	Day 7		Day 14		Day 28	
	Size (nm) ± SD	PDI ± SD	Size (nm) ± SD	PDI ± SD	Size (nm) ± SD	PDI ± SD
**ASO-Li-1**	150.7 ± 6.1	0.16 ± 0.03	not performed			
**ASO-iLi-1**	166.9± 6.4	0.18 ± 0.07	not performed			
**ASO-Li-2**	148.3 ± 8.4	0.07 ± 0.01	148.2 ± 5.3	0.08 ± 0.03	150.4 ± 4.7	0.11 ± 0.05
**ASO-iLi-2**	151.6 ± 1.0	0.06 ± 0.01	154.4 ± 3.4	0.08 ± 0.03	162.3 ± 1.0	0.10 ± 0.03

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
