# Peer review of "Enhanced Antisense Oligonucleotide Delivery Using Cationic Liposomes Grafted with Trastuzumab: A Proof-of-Concept Study in Prostate Cancer"

_pharmaceutics, 2020, doi:10.3390/pharmaceutics12121166_

Round 1
Reviewer 1 Report
Manuscript ID: pharmaceutics-1018568
The manuscript of Guillaume Sicard, Clément Paris, Sarah Giacometti, Anne Rodallec, Joseph Ciccolini, Palma Rocchi, and Raphaëlle Fanciullino as Co-authors: “Enhanced Antisense Oligonucleotide Delivery Using Cationic Liposomes Grafted with Trastuzumab: A proof-of-concept study in prostate cancer” presents the studies regarding delivery of antisense oligonucleotide using cationic liposomes. This manuscript was resubmitted a new version, main suggestions of reviewer are included in the manuscript.
However, there are some additional issues needed to be addressed before publication:
- Please, use the numbers of Figures in ascending order. Authors used two times Figure 6 and divided information in the two parts.
- Line 235, Fig. 4, please check the quality of text into marked line.
- Line 285: check style points for the legend of Fig. 10.
- Table 1. Please use the same style point and use the same significant digits for all data.
- Please add in Table 2 also data regarding formulation-1.
- Please use the same style point for the description of Trastuzumab, sometimes authors use Trastuzumab, sometimes - trastuzumab,
- Please rewrite conclusions giving more detailed information and conclusion regarding your studies. Now only general information is given in conclusions.
Consequently, I do recommend accepting this manuscript for publication with minor revision.
Author Response
Dear,
All requested modifications have been made in order to respond to your requests.
Best regards,
Raphaëlle Fanciullino

Reviewer 2 Report
The authors have fulfilled most of the suggestions pointed out in the earlier version. the manuscript may be accepted for publication.
Author Response
Dear,
Would like to thank you for your review.
Best regards
Raphaëlle Fanciullino